# Excellent Electrochromic Properties of Ti^4+^-Induced Nanowires V_2_O_5_ Films

**DOI:** 10.3390/ma17194680

**Published:** 2024-09-24

**Authors:** Yufei Deng, Hua Li, Jian Liang, Jun Liao, Min Huang, Rui Chen, Yinggui Long, Jacques Robichaud, Yahia Djaoued

**Affiliations:** 1Department of Materials Chemistry, School of Materials Science and Engineering, Jingdezhen Ceramic University, Jingdezhen 333403, China; 2Laboratoire de Recherche en Matériaux et Micro-Spectroscopies Raman et FTIR, Université de Moncton-Campus de Shippagan, Shippagan, NB E8S 1P6, Canada; 3National Engineering Research Centre for Domestic & Building Ceramics, Jingdezhen Ceramic University, Jingdezhen 333001, China

**Keywords:** V_2_O_5_, Ti ion, nanowire, electrochromic, cycling stability

## Abstract

Ti^4+^-doped V_2_O_5_ films with nanowires on top and a dense, long nanorod layer on the bottom were successfully fabricated using the spin-coating route. During the electrochromic cycling, charge transfer resistance (*R_ct_*) decreases while ion-diffusion ability (*K_Ω_*) rapidly drops in the first ten cycles and then levels off. Low *R_ct_* and morphology of nanowires collaboratively improved the electrochromic behavior of Ti^4+^-doped V_2_O_5_ films by enhancing the charge transfer speed and minimizing polarization and dissolution. The obtained Ti^4+^-doped V_2_O_5_ film shows better electrochromic properties than the undoped V_2_O_5_ film, with a coloration efficiency (CE) of 34.15 cm^2^/C, coloration time of 9.00 s, and cyclic retention of 82.6% at cycle 100. In contrast, the corresponding values for the undoped V_2_O_5_ film were 23.57 cm^2^/C, 13.16 s, and 43.6%.

## 1. Introduction

Transition metal oxides (TMOs) offer a wide range of applications for energy storage and energy conversion [1], waste treatment [2], and gas sensing, to name a few. For instance, Ishibe et al. showed the power enhancement of embedded-ZnO nanowire structures for energy conversion [3], while Maeng et al. presented a highly sensitive SnO_2_ nano slab for NO_2_ gas sensing [4].

Among TMOs, V_2_O_5_, owing to its layered structure and high lithium-ion intercalation capacity, has attracted considerable attention for promising applications in devices such as batteries [5,6], supercapacitors [7], and electrochromic (EC) smart windows [8]. However, the practical applications of V_2_O_5_ in these fields are limited due to its poor cycling stability [9,10], which is generally attributed to low electronic conductivity and/or ionic conductivity [11,12], slow ion diffusion [13], and inert fragile structure [14]. 

Aiming to increase conductivity, some strategies have been adopted, such as incorporating a highly conductive material, introducing low valence or non-stoichiometric vanadium oxide [8], introducing a nanostructure or porous structure design, doping, or adding a small amount of another oxide [15]. On one hand, conductive materials, such as carbonaceous materials [16], conductive polymers [17], and metal oxides [18], significantly improve cycling stability by reducing the coexistence of multiple phases and dissolution of soluble intermediates in these multiple phases through fast charge transfer speeds [19]. On the other hand, they often decrease the visible transmittance region, which is inappropriate for EC smart window applications. The same issue occurs when introducing low valence vanadium oxide. Apart from incorporating a highly electrically conductive material, nanostructured materials and porous structures have also interested researchers because of their large surface area and corresponding shortened diffusion pathways for Li^+^ [20]. Various nanostructures have been prepared, such as nanorods [20], nanobelts [21], and even hierarchical nanostructures [22]. Porous structures [23,24], with an interconnected network of nanometer-thick walls suitable for electrolyte penetration and giving effective charge/ions transport pathways, were also prepared and deeply explored. Porous and nanostructured V_2_O_5_ suffer from poor stability due to their intrinsic low conductivity. However, by compositing V_2_O_5_ with other metal oxides [15,25,26] or doping with heteroatoms such as Y [27], Cu [28], Sn [29], etc., capacity retention has been greatly improved. The improvement was attributed to low impedance and, therefore, smaller polarization and faster kinetics [27]. 

Other than cycling stability, switching time, coloration efficiency (CE), and optical contrast are also critical parameters in an EC device (ECD). Switching time depends on several factors, including electronic conductivity, ion diffusion ability, the magnitude of the applied potential, morphology, and so forth [30,31]. Coloration efficiency is contingent upon the specific type of EC material, the quantity of charge insertion, and the voltage program [31]. In contrast to its application in batteries, the coloration mechanism of V_2_O_5_ in EC devices is far less explored and yet to be clearly understood. Furthermore, the present works on V_2_O_5_ mainly focus on exploring the effect of initial impedance on cycling stability, presuming that the impedance is constant during cycling. However, this is not the case since chemical dissolution and structural fracture continuously occur during cycling. 

Herein, we first prepared Ti^4+^-doped V_2_O_5_ films with nanowire morphology on top and a dense, long nanorod layer on the bottom using the spin-coating route followed by annealing in the air. To our knowledge, this is the first time that the collation between EC properties and impedance of Ti-induced nanowire film has been dynamically investigated upon cycling. This gives us a better understanding of decaying upon cycling and, therefore, benefits the potential practical application of V_2_O_5_-based films. The EC properties upon cycling, including coloration time and cycling stability, were evaluated.

## 2. Materials and Methods

### 2.1. Materials

Vanadium oxytripropoxide (OV(OC_3_H_7_)_3_, VTIP) (98%), titanium tetraisopropoxide (Ti(OC_3_H_7_)_4_, TTIP) (97%), and lithium perchlorate (LiClO_4_) (99%) were purchased from Sigma-Aldrich LLC (St. Louis, MO, USA). Isopropanol (C_3_H_7_OH) (99%), acetylacetone (C_5_H_8_O_2_) (99%), acetic acid (CH_3_-CO_2_H) (99%), and ethanol (C_2_H_5_OH) (99.7%) were purchased from Sinoreagent Co. Ltd. (Beijing, China). Propylene carbonate (C_4_H_6_O_3_) (99%) was purchased from Ourchem LLC. (Guangzhou, China), and Triton X-100 (C_34_H_62_O_11_) was purchased from Macklin LLC (Albany, NY, USA).

### 2.2. Synthesis of V_2_O_5_ Precursor Solution

The V_2_O_5_ precursor solution was obtained by dissolving 0.3 mL vanadium oxytripropoxide into 9.98 mL isopropanol, which was added into an ethanolic solution of acetic acid (0.012 mL acetic acid in 9.98 mL ethanol). Then, 1 mL Triton X-100 was added to the obtained solution and stirred for 1.5 h.

### 2.3. Synthesis of Ti^4+^-Doped V_2_O_5_ Precursor Solution

First, solution A was prepared by adding 0.6 mL titanium tetraisopropoxide into a 12 μL acetylacetone in a 7.9 mL isopropanol solution under continuous stirring until a clear transparent solution was formed. Second, solution B was obtained by introducing a 0.3 mL vanadium oxytripropoxide solution in 9.98 mL isopropanol into a mixture of 0.012 mL acetic acid with 9.98 mL ethanol. Then, 0.195 mL of solution A was added gradually, in drops, to solution B until the solution cleared. The final solution was obtained by adding 1 mL Triton X-100 to the mixture of solutions A and B and stirring for another 1.5 h.

### 2.4. Fabrication of Undoped V_2_O_5_ and Ti^4+^-Doped V_2_O_5_ Films

The films were formed on transparent indium tin oxide (ITO)-conducting glass substrates by using spin-coating. The precursor solutions (36 μL drops) were spread onto 2.5 cm × 2.5 cm ITO substrates and allowed to uniformly cover the substrates for 30 s before starting the spin-coating process for 30 s at a spin rate of 1000 revolutions per minute (rpm). The spin-coating process was repeated twice to obtain three layers on the ITO glass. A sufficient time interval (30 s) was provided for air-drying between successive coatings.

All films were further calcined at 450 °C for two hours. Thus, the films obtained were V_2_O_5_ film and Ti^4+^-doped V_2_O_5_ film, respectively. 

### 2.5. Fabrication of EC Devices

The EC devices were constructed with the following configuration: ITO-coated-glass-1/Ti^4+^-doped V_2_O_5_ film (or V_2_O_5_ film)/electrolyte/ITO-coated-glass-2, where ITO-coated-glass-1 and ITO-coated-glass-2 are the two transparent electrodes (TEs) used to apply the electric field, and Ti^4+^-doped V_2_O_5_ film (or V_2_O_5_ film) is the EC layer. The electrolyte was 0.5 mol/L lithium perchlorate in propylene carbonate. After making the electrical connections, the EC device was ready for testing. The area of the EC device was 2.0 cm × 2.0 cm.

### 2.6. Material Characterization

The films’ morphology was characterized using a field emission scanning electron microscope (Hitachi S-4800 FE-SEM, Ibaraki-ken, Japan). The thickness of the films was measured from their cross-sectional SEM images using ImageJ 1.8.0 software. For phase analysis, Raman spectroscopy was recorded at room temperature in the wavenumber range from 50 to 1200 cm^−1^ at an excitation wavelength of 532 nm (Thermo Scientific, Waltham, MA, USA). The spectra were generated with ~0.45 mW, 633 nm He-Ne laser excitation at the sample surface. Transmission electron microscope (TEM) images were obtained on a JEOL-2010F electron microscope operated at 200 kV.

### 2.7. Electrochemical and EC Measurements

Electrochemical impedance spectroscopy (EIS) and cyclic voltammetry (CV) were carried out on an electrochemical workstation CHI600E (Chinstruments, Shanghai, China) using a three-electrode cell at open circuit voltage. EIS and CV measurements used the Ti^4+^-doped V_2_O_5_ film (or V_2_O_5_ film) deposited on ITO substrates as working electrodes. At the same time, a platinum grid served as a counter electrode, and a commercial Ag/AgCl 1 M KCl electrode served as a reference. A 0.5 mol/L LiClO_4_/propylene carbonate solution was used as an electrolyte. The test parameters of EIS were measured in the frequency range from 0.01 Hz to 1000 kHz at the open circuit voltage (OCV) after discharge/charge cycles of 5 mV amplitude. CV measurements were performed in the voltage range from −1.5 to 1.5 V at a cycling speed of 100 mV/s. 

EC measurements on the fabricated EC devices were conducted by combining optical transmittance spectra using a UV-3600 spectrophotometer in the wavelength range of 190–1100 nm (Shimadzu, Tokyo, Japan) with chronoamperometry technology (CA) at an applied voltage from −3.0 V to +3.0 V, in increasing potential steps of 0.5 V kept for 30 s. 

The corresponding change in optical density (Δ*OD*) was defined as the following [32]:(1)$$\Delta \mathrm{OD}=\log(T_{bleached}/T_{colored})$$

Coloration switching time (*τ_c_*) is the time required to reach 90% of the film’s complete transmittance change. The films’ coloration switching behavior was measured at wavelengths of 400 nm (*τ_c400_*), while cycling stabilities were measured at a wavelength of 400 nm. A square wave voltage of +3.0 V to −3.0 V was alternately applied for 30 s for each state and was kept for 100 cycles. 

Cycling stability was characterized by retention. *Retention* was calculated as the following [33]:(2)$$Retention=\frac{\Delta T_{n}}{\Delta T_{1}}$$
where $\Delta T_{1}$ is the optical contrast of the 1st cycle, and $\Delta T_{n}$ is the optical contrast of the *n^th^* cycle. 

Coloration efficiency (*η*) was extracted as the slope of the line fitted to the linear region of the curve of OD versus the extracted charge density at −3.0 V of coloring potential at 400 nm [9] and expressed as the following [34]:(3)$$\eta =\frac{\Delta OD}{q}$$

## 3. Results and Discussion

### 3.1. Microstructural Characterization

Figure 1a shows the Raman spectra of V_2_O_5_ and Ti^4+^-doped V_2_O_5_ films. The distinctive Raman modes of α-V_2_O_5_ emerge at ~100, 142, 196, 281, 301, 405, 482, 526, 700, and 992 cm^−1^. The low-frequency modes at ~100, 142, and 196 cm^−1^ correspond to the relative motions of V_2_O_5_ layers (external modes) in V_2_O_5_ [8]. The two peaks at 142 and 196 cm^−1^ are intimately linked with the layered structure, proving long-range structural order [34]. The intermediate frequency peaks at ~281, 301, 405, 482, 526, and 700 cm^−1^ are related to the bending and stretching vibrations (internal modes) of the vanadium-oxide bond in V_2_O_5_ [35]. The peak at 281 cm^−1^ is attributed to the V = O bending vibration, peaks at 301 and 700 cm^−1^ correspond to the V_3_–O (triply coordinated oxygen) stretching mode, and peaks at 482 and 526 cm^−1^ correspond to the bending vibrations of the V–O–V (bridging doubly coordinated oxygen) bending vibration. The highest frequency peak at ~992 cm^−1^ corresponds to the stretching mode of the terminal oxygen (vanadyl oxygen, V = O_ν_) [36].

Despite that, the Raman modes of the anatase TiO_2_ phase, which should appear at ~145 and 198 cm^−1^, are close to those of α-V_2_O_5_ at 142 and 196 cm^−1^; the other Raman modes of the anatase TiO_2_ phase, at ~396, 515, and 640 cm^−1^, are not observed in the spectrum of the Ti^4+^-doped V_2_O_5_ film, demonstrating that there is no isolated TiO_2_ phase in the Ti^4+^-doped film. The doping process could be represented by Equation (4).
(4)$${TiO}_{2}\overset{V_{2}O_{5}}{\to }2{Ti}_{V}^{\prime }+V_{O}^{\cdot \cdot }+2O_{O}$$

In Figure 2, the SEM image of the undoped V_2_O_5_ film displays many tiny rod-like particles with ~200 nm length and ~100 nm diameter, aggregating to form a dense porous film (Figure 2a,e). In contrast, in the SEM image of the Ti^4+^-doped V_2_O_5_ film (Figure 2b), large nanowires are seen on top, as illustrated by green arrows, and long nanorods at the bottom, red arrows. The nanowires (top layer) have nearly the same diameter (~60 nm), with a length ranging from several hundred nanometers to tens of micrometers. Different from the rod-like particles in the undoped V_2_O_5_ film, the long nanorods on the bottom of the Ti^4+^-doped V_2_O_5_ film interlace to form a dense layer. Furthermore, the nanowires on the top seem to grow up from the bottom part with one end inserted within the bottom layer, indicating good interconnection between the top nanowires and the bottom long nanorods layer. The cross-sectional SEM images demonstrate that several nanowires grow from the bottom of undoped V_2_O_5_ film (see inset of Figure 2a), with the thickness of the bottom part at around 110 nm. After doping with Ti^4+^, the nanowires became dense, with a thickness of around 130 nm (inset of Figure 2b). Therefore, it is reasonable to deduce that the formation of nanowires comes from introducing Ti ions. Corresponding EDS results (Figure 2g) show that the atomic ratio Ti:V for the Ti^4+^-doped V_2_O_5_ film is 0.03:1. 

Figure 3a,b shows the TEM images of a single Ti^4+^-doped V_2_O_5_ nanowire, whereas Figure 3c is the HRTEM image taken from the square frame shown in Figure 2a along with its diffraction pattern as an inset. The (101) interplanar spacing of Ti^4+^-doped V_2_O_5_ (Figure 3c) is enlarged (3.47 Å) in contrast to that of undoped α-V_2_O_5_, (3.20 Å), based on PDF#53-0538 [37]. Such an enlarged lattice could be attributed to the larger size of Ti^4+^ ions (0.61 Å) compared to V^5+^ ions (0.54 Å). Therefore, by combining Raman and SEM results, it was deduced that Ti^4+^ ions had been successfully inserted into the V_2_O_5_ lattice. Figure 3d shows the EDS retention mapping of all atoms in the Ti^4+^-doped V_2_O_5_ nanowire, and Figure 3e–g shows the atomic distribution of V, Ti, and O, respectively. Furthermore, this EDS mapping demonstrates that the Ti ions are uniformly distributed inside the Ti^4+^-doped V_2_O_5_ nanowire (Figure 3f). Additionally, the Ti:V ratio from the EDS spectrum obtained from TEM (Figure 3h) is 0.03:1.

EIS was used to understand and analyze the kinetic behavior and calculate the values of the circuit components of Li^+^ ion intercalation/deintercalation during cycling. The depressed semicircle in Nyquist plots (Figure 4a,b) observed in the high-frequency region is due to the charge transfer resistance (*R_ct_*). In contrast, the inclined line in the low-frequency region reflects Warburg impedance [38]. The films show a capacitive arc in the high-frequency region and diffusion in the low-frequency domain. The *R_ct_* values were determined by plot fitting using ZView2 software with the equivalent circuit depicted in Figure 4a,b insets. The slope of the inclined line in the low-frequency region after the semi-circle is related to Li-ion diffusion and is named *K_Ω_*, which is the ion diffusion ability. Generally, a high *K_Ω_* corresponds to fast ion diffusion or speed [14,39].

As seen in Figure 4c, *R_ct_* values decrease almost linearly with cycling for both undoped V_2_O_5_ and Ti^4+^-doped V_2_O_5_ films. The decrease is similar for both films since the slopes of *R_ct_* vs. the number of cycles are almost the same. The reduction in *R_ct_* could originate from vanadates, such as H_2_VO_4_^−^, HVO_4_^2−^, HV_2_O_5_^−^, and vanadium-bronze species formed during Li-ion intercalation/deintercalation [19]. The formed vanadates dissolve into the electrolyte, leading to a decrease in the amount of V_2_O_5_ and therefore to a decrease in *R_ct_*. Further, the formed vanadium-bronze remains within the V_2_O_5_ lattice, decreasing *R_ct_* [40]. In addition, throughout the cycling process, the Ti^4+^-doped V_2_O_5_ film showed much smaller *R_ct_* values than those of the undoped V_2_O_5_ film (Figure 4c), indicating faster charge transfer in the Ti^4+^-doped V_2_O_5_ film. Such an observation is consistent with results from M. Panagopoulou et al., in which the charge transfer resistance decreases after doping Mg^2+^ into V_2_O_5_ [15]. However, the *K_Ω_* values for both samples are nearly the same during the cycling (Figure 4d), indicating that doping V_2_O_5_ with Ti ions does not improve the ion diffusion speed. Furthermore, a vast drop occurs from cycle 0 to cycle 10 in both samples, after which *K_Ω_* values remain constant, indicating that a dramatic structure change happens from the beginning of Li-ion intercalation/deintercalation; afterward, the intercalation/deintercalation process becomes dynamically balanced without further change upon cycling. 

### 3.2. EC Properties

Figure 5a,b show the transmittance spectra of the EC device constructed using undoped and Ti^4+^-doped V_2_O_5_ films, respectively. After coloration, the transmittance of both samples increases in the visible region while it decreases in the NIR region. For instance, for the Ti^4+^-doped V_2_O_5_ film, the transmittance at 400 nm in the colored state (*T_c_*) at −3.0 V is 35.90%, which is higher than that of the as-prepared state at 0 V (T_0_ = 17.8%) (Table 1), while at 900 nm, it is 56.30%, which is lower than the 71.80% at 0 V. On the other hand, the Ti^4+^-doped V_2_O_5_ film has better reversibility than the undoped V_2_O_5_ film by showing a spectrum at the bleached state (+3 V) closer to that of the as-prepared state (0 V). 

Here, reversibility was calculated as the following:(5)$$\mathrm{Reversibility}=100\%\times (1-\frac{\left|T_{0}-T_{b}\right|}{T_{0}})$$
where *T*_0_ and *T_b_* are the transmittances at 0 V (as prepared state) and +3 V (bleached state).

The calculated reversibility at 400 nm for Ti^4+^-doped V_2_O_5_ film is 100%, noticeably better than that of the undoped V_2_O_5_ film, which agrees with the smaller *R_ct_* of the Ti^4+^-doped V_2_O_5_ film.

The absolute value of optical density (|Δ*OD*|) for Ti^4+^-doped V_2_O_5_ film is much higher than that of the undoped V_2_O_5_ film in the range from 320 to 1100 nm (Figure 5c). For instance, |Δ*OD*| for the Ti^4+^-doped V_2_O_5_ film is 0.30 at a wavelength of 400 nm, which is almost twice that of the undoped V_2_O_5_ film (0.19) (Table 1). Considering that the largest |Δ*OD*| occurs at a wavelength of ~400 nm for both samples, the following tests for the EC properties, including switching time, coloration efficiency, and cycling stability, were conducted at 400 nm to achieve a strong response.

Coloration efficiency (CE), which is defined as the change in optical density (*OD*) per unit of charge (*Q*) intercalated into or extracted from the EC film (see Equation (4)) [41,42], was found by calculating the slopes of Δ*OD* versus Δ*Q* at 400 nm, as seen in Figure 5d,e. CE is a practical parameter to measure the power requirements for the color-switching process, and a higher CE means better electronic utilization efficiency [31]. Coloration efficiency depends on the kind of EC material, amount of charge insertion, and voltage programs [31]. However, the physical basis for coloration is yet to be understood entirely [43,44]. In this work, the undoped V_2_O_5_ film has a CE of 23.57 cm^2^/C, close to a value reported by Liu et al. (24.5 cm^2^/C) [44]. After doping, the Ti^4+^-doped V_2_O_5_ film shows an improved CE of 34.15 cm^2^/C. This value is even higher than those reported in the literature, in which it is 4.7 cm^2^/C for V_2_O_5_-TiO_2_ coating with V/Ti (70/30) [25], 18.6 cm^2^/C for RF magnetron sputtered Mg^2+^-doped V_2_O_5_ film [15], and 24.12 cm^2^/C for 3.5:1 (*V*/*W*) composite W/WO_3_-V_2_O_5_ films [26]. Considering that both undoped and Ti^4+^-doped V_2_O_5_ films have close electrochemical capacities, as discussed above, such an excellent CE value (34.15 cm^2^/C) of Ti^4+^-doped V_2_O_5_ film could be attributed to the lower *R_ct_*, which renders faster Li^+^ intercalation/deintercalation and also faster charge transfer speed, and, therefore, better charge utilization efficiency than the undoped V_2_O_5_ film [19]. 

For the Ti^4+^-doped V_2_O_5_ film, coloration time (*τ_c_*) first increases rapidly with cycling, peaking at 16.10 s for cycle 20 (Figure 6), after which it decreases, reaching ~9.00 s at cycle 100. For the undoped V_2_O_5_ film, *τ_c_* also rapidly increases with cycling at almost the same rate as the Ti^4+^-doped V_2_O_5_ film until it reaches around cycle 10. From then on, its quick increase stops, and *τ_c_* continues to grow at a somewhat constant rate until it reaches 13.16 s at cycle 100. At cycle 60, both samples have a close *τ_c_* (12.19 s for the undoped V_2_O_5_ film and 12.65 s for the Ti^4+^-doped V_2_O_5_ film). After cycle 60, the Ti^4+^-doped V_2_O_5_ film shows shorter coloration times than the undoped V_2_O_5_ film.

Many factors can affect switching speed, including electronic conductivity, electrode materials, the underlying conductive layers, the ionic conductivity of the electrolyte, the morphology of the EC layer, the associated changes in ion diffusion within this morphology of the EC layer, and ion insertion kinetics [30,39]. Previous studies have shown the positive direct effect of the electronic conductivity of the electrode materials on coloration time [23,31]. However, in the present work, other factors influence coloration time because the Ti^4+^-doped V_2_O_5_ film does not show a shorter coloration time than the undoped V_2_O_5_ film during the cycling process despite its lower *R_ct_* and their close *K_Ω_* values.

Undoped and Ti^4+^-doped V_2_O_5_ films have different morphologies, as is seen in SEM images (Figure 2a,b). Apart from nanowires on the top layer, the Ti^4+^-doped V_2_O_5_ film has a dense bottom layer, while the undoped V_2_O_5_ film features a developed porous structure. The developed porous structure in the undoped film and its corresponding rough surface provide the advantages of quickly absorbing the electrolyte and favoring Li^+^ ion intercalation compared to the dense Ti^4+^-doped V_2_O_5_ film. Therefore, in the first five cycles, as potential was just applied to the samples, the undoped film shows a close coloration time to the Ti^4+^-doped film despite its lower *R_ct_*. Such an increase in both undoped and Ti^4+^-doped V_2_O_5_ films indicates that at the beginning of the EC process, an activation is needed to adjust the interaction between the electrolyte and electrode. After five cycles, the amount of Li^+^ ions intercalated almost reached saturation, combining a stable structure after the primary activation, and the coloration time for the undoped film became nearly leveled. As for the Ti^4+^-doped film, such a saturation process occurs at cycle 20, at which point the coloration time peaks. After cycle 20, the nanowire morphology on the top of the Ti^4+^-doped film reveals its importance by sharing a relatively high-strength electric field at its sharp edges [23], rendering a weaker electric field on the dense bottom layer. The polarization is significantly reduced since the high electric field was removed, facilitating Li^+^ ion intercalation. In addition to its reduced *R_ct_* with cycling, coloration time quickly decreases from cycle 20 to 100.

Cycling stability was investigated at 400 nm, defined as the number of cycles required for retention to decline to 60% of its value at cycle 1. In the first five cycles, retention of the undoped V_2_O_5_ film underwent a nearly vertical drop to 79.7% (Figure 7c) and then decreased slowly at a constant rate, reaching 43.6% at cycle 100. As for the Ti^4+^-doped V_2_O_5_ film, retention initially suffered a fast decrease, though not as steep as for the undoped V_2_O_5_ film. However, a turning point occurs at cycle 20, reaching the minimum retention of 77.39%, which coincides with the cycle at which coloration time is at a maximum (Figure 7c). 

After cycle 20, retention levels off (~81%) with slight variations until cycle 100, for which retention is 82.6%. The experimental results we have obtained are significant. Our tested material outperforms those from the literature, which report 65% after 600 cycles for a 4 at.% Ti^4+^-doped V_2_O_5_ planar film [9] and 50% after 100 cycles for an 8.5 at.% Ti^4+^-doped V_2_O_5_ film [45], 76% after 100 cycles for a 5% at.% Ti^4+^-doped V_2_O_5_ film [46], 18.6% for a nanobelt-membrane hybrid-structured vanadium oxide film [21], and 24% for a self-organized multifunctional V_2_O_5_ nanofiber-liquid crystal polymer hybrid film [47].

In the present work, the fast fading in the first several cycles for both undoped and doped samples indicates that an important irreversible structural change and/or chemical dissolution primarily occurred at the beginning of cycling, which aligns with the continuous drop in R_ct_. In addition, faster fading for the undoped film compared to the Ti^4+^-doped film is observed at the beginning of the cycling process. As discussed, regarding coloration time, a highly developed porous structure plays a double-edged role throughout process: On one hand, it facilitates the electrolyte absorption and Li^+^ ion intercalation; on the other hand, such a rapid electrochemical process accelerates irreversible structural change and/or chemical dissolution. In addition to a larger R_ct_ value of the undoped film than that of the Ti^4+^-doped film, the faster chemical dissolution leads to its faster fading [17]. Apart from this, a dark gray layer remained in the bleached state after cycle 1, as seen in Figure 7a, indicating a failed delithiation and the remaining V^4+^ compounds. These V^4+^ compounds slowly dissolve into the electrolyte during the cycling process because, at cycle 100, the dark layer becomes paler (Figure 7a). In contrast, there is no formation of a dark layer in Ti^4+^-doped V_2_O_5_ film in the bleaching state after cycle 1 (Figure 7b, bleaching) due to its low charge transfer resistance and reduced V^4+^ compound formation. A strong fading occurs in the bleaching state after the first cycle in both samples since their color became paler than the as-prepared samples, as seen in Figure 7. After 100 cycles, the Ti^4+^-doped V_2_O_5_ film kept nearly the same color as in cycle 1, while the undoped V_2_O_5_ film became paler than in cycle 1, consistent with the retention results (Figure 7c).

Apart from the low charge transfer resistance, the nanowires in the Ti^4+^-doped V_2_O_5_ film play a primary role in cycling stability since their nano edges reduce polarization through a shared high-strength electric field and, therefore, minimize those dissolvable intermediate products. The SEM images in Figure 2c,d further illustrate this observation. After 100 cycles, the undoped V_2_O_5_ film is transformed entirely and mostly dissolved, leaving only some scattered nanoparticles on the surface, highlighted with yellow circles in Figure 2c. In contrast, the Ti^4+^-doped V_2_O_5_ film kept its original morphology with nanowires on top and nanorods on the bottom layer. The corresponding EDS spectrum (Figure 3h) shows that, after 100 cycles, the atomic ratio Ti:V in the Ti^4+^-doped V_2_O_5_ film is around 0.02:1, close to that of its as-prepared state (0.03:1). In contrast, for the undoped V_2_O_5_ film, the content of V ions is too low to be detected in the EDS spectrum (highlighted with red circles in Figure 2g), indicating that most of the V_2_O_5_ was lost, resulting in correspondingly high fading upon cycling. Raman results (Figure 1b) show that after 100 cycles, both V_2_O_5_ and other vanadium oxide phases are formed in the Ti^4+^-doped V_2_O_5_ film, while there is no V_2_O_5_ phase found in what was originally the undoped V_2_O_5_ film. Doping V_2_O_5_ with a small amount of Ti ions transformed the morphology of the film’s top surface by forming a layer of long nanowires, which enhanced the conductivity of the film and improved its cycling stability.

## 4. Conclusions

Ti^4+^-doped V_2_O_5_ films were successfully fabricated by spin-coating, featuring nanowires on top and a dense long nanorod layer on the bottom. The nanowires show a uniform diameter of around ~60 nm, with lengths from several hundred nanometers to tens of micrometers. After doping, the (110) interplanar lattice spacing of the undoped V_2_O_5_ film enlarged to 3.47 Å from 3.20 Å. Upon EC cycling, a dynamic EIS characterization was used to analyze the EC properties, including coloration time, and cycling stability. During the cycling process of both undoped and doped samples, *R_ct_* decreased with cycling while *K_Ω_* rapidly dropped in the first 10 cycles and then leveled off, independent of cycling. However, the Ti^4+^-doped V_2_O_5_ film showed a much lower *R_ct_* than the undoped film. The introduction of Ti ions significantly influences EC properties in two ways. First, it lowered the *R_ct_* value; and second, it triggered the formation of nanowires. A small *R_ct_* value, corresponding to fast charge transfer speeds, delivers high CE, fast coloration time, and, most importantly, better cycling stability. The nanowires, taking advantage of their sharp nano-edges to share a relatively high-strength electric field, improve coloration speed and cycling stability by minimizing polarization and the corresponding dissolution of the formed intermediate phases. With a CE of 34.15 cm^2^/C, a coloration time of 9.00 s at cycle 100, and a cyclic retention of 82.6%, the thus obtained Ti^4+^-doped V_2_O_5_ film showed better EC properties than the undoped V_2_O_5_ film, for which those values are 23.57 cm^2^/C, 13.16 s, and 43.6%. The paper’s significance lies in the new insight into the effect of morphological and structural properties.

## Figures and Tables

**Figure 1 materials-17-04680-f001:**
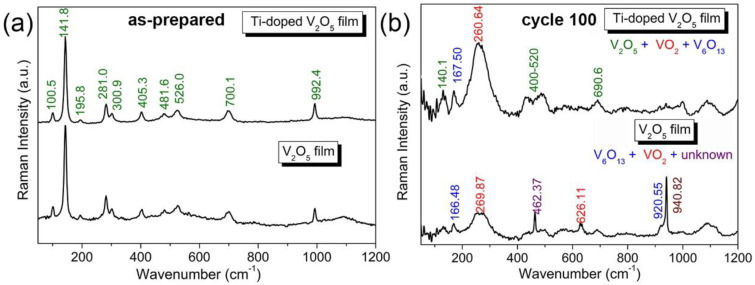
Raman spectra for V_2_O_5_ and Ti^4+^-doped V_2_O_5_ films: (**a**) before cycling and (**b**) after cycle 100.

**Figure 2 materials-17-04680-f002:**
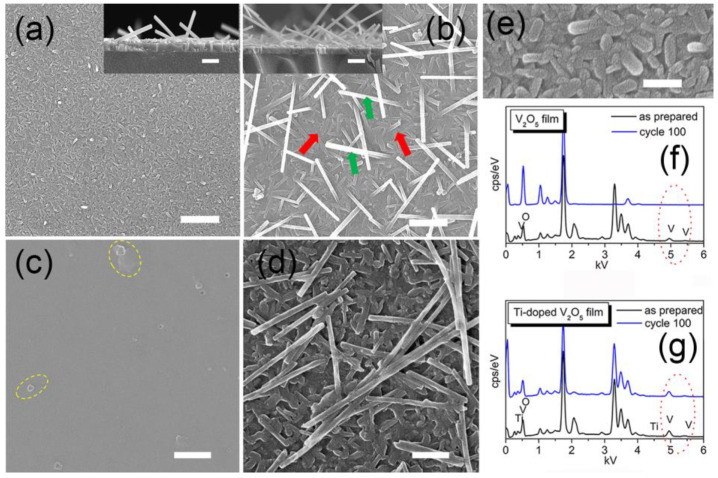
SEM images of: (**a**,**c**,**e**) V_2_O_5_ film; (**b**,**d**) Ti^4+^-doped V_2_O_5_ film (nanowires (top) indicated with green arrows and the nanorod layer (bottom) with red arrows). (**f**,**g**) Corresponding EDS results for the films in the as-prepared state (**a**,**b**,**e**) and after 100 EC cycles (**c**,**d**). Inset in (**a**,**b**) showing the corresponding thickness of the films. Scale bars in Figures (**a**,**b**) 1 μm, in Figures (**c**,**d**) 500 nm, in Figure (**e**) 200 nm, and insets in Figures (**a**,**b**) 500 nm. Yellow circles in Figure (**c**) indicate nanoparticles left after cycling, and red circles in Figures (**f**,**g**) highlight the content changes of V ions before (black spectrum) and after (blue spectrum) cycling.

**Figure 3 materials-17-04680-f003:**
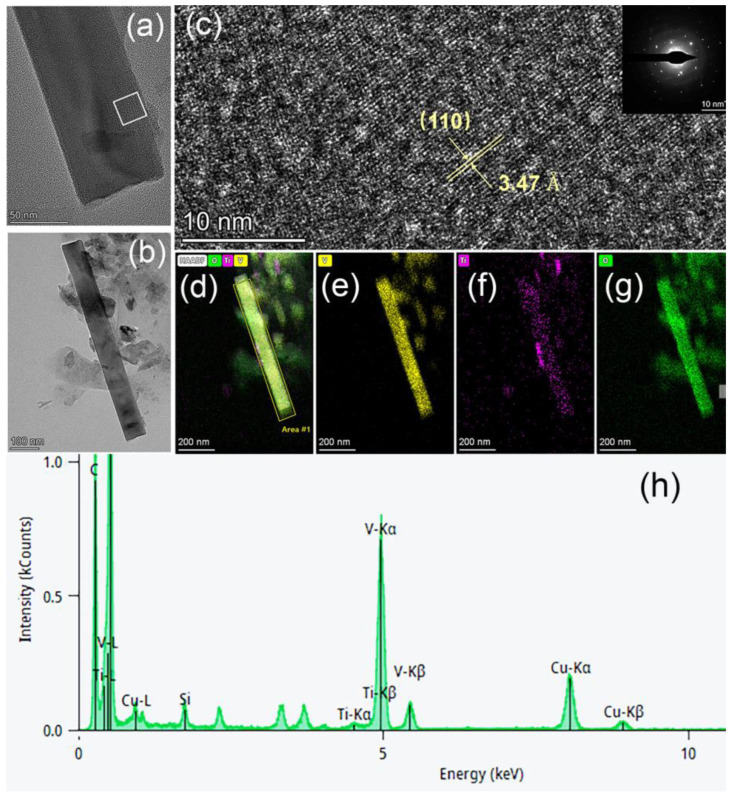
TEM images (**a**,**b**) of a single Ti^4+^-doped V_2_O_5_ nanowire. (**c**) HRTEM image taken from the square frame in (**a**) and its diffraction pattern (inset). (**d**) Retention mapping showing all atoms. Atomic distribution of (**e**) V, (**f**) Ti, and (**g**) O. (**h**) Corresponding EDS result for the films.

**Figure 4 materials-17-04680-f004:**
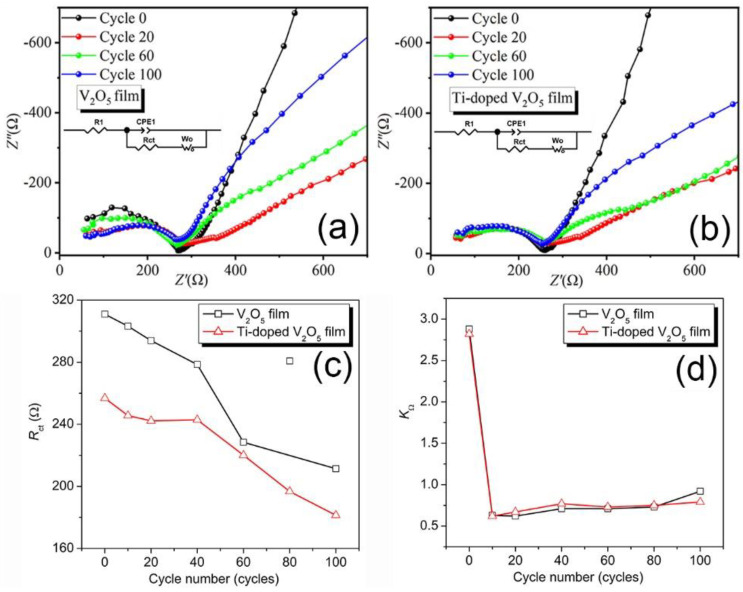
Nyquist plots of V_2_O_5_ film (**a**) and Ti^4+^-doped V_2_O_5_ film (**b**) at cycle 0 (black), cycle 20 (red), cycle 60 (green), and cycle 100 (blue), with equivalent circuits shown as insets. Evolution of *R_ct_* (**c**) and *K_Ω_* (**d**) as a function of cycling for V_2_O_5_ film (square, black) and Ti^4+^-doped V_2_O_5_ film (triangle, red).

**Figure 5 materials-17-04680-f005:**
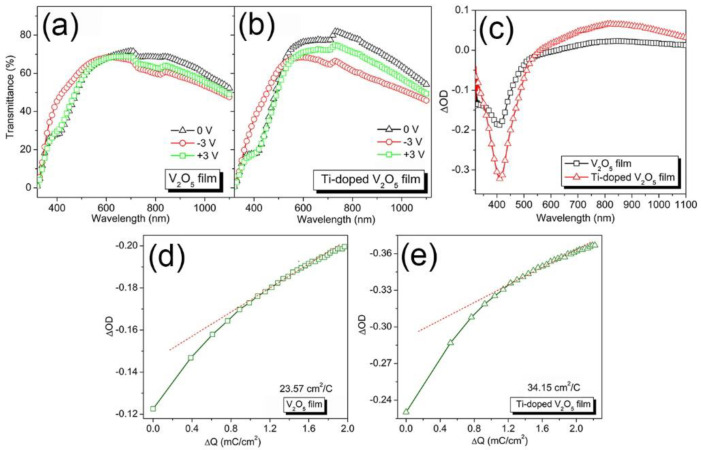
Transmittance spectra of the EC device constructed using (**a**) the undoped V_2_O_5_ film and (**b**) the Ti^4+^-doped V_2_O_5_ film in the as-prepared state 0 V (black, triangle), at a coloration potential of −3.0 V (red, sphere), and a bleaching potential of +3.0 V (green, square); (**c**) optical density; (**d**,**e**) coloration efficiency taken from the fifth cycle, in which the red line indicates the line fitted to the linear region of the curve (green) of OD versus ΔQ.

**Figure 6 materials-17-04680-f006:**
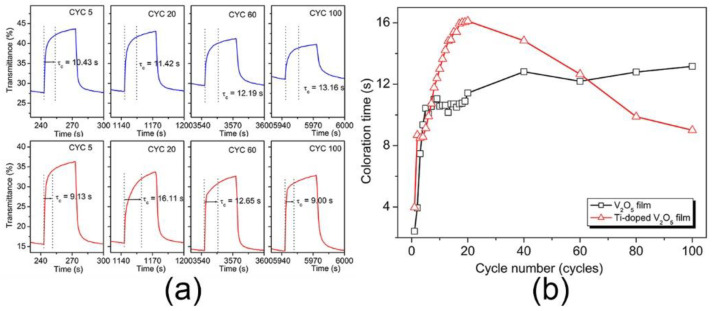
(**a**) Coloration switching time of V_2_O_5_ film (top) and Ti^4+^-doped V_2_O_5_ film (bottom) at a wavelength of 400 nm after 5, 20, 60, and 100 cycles; (**b**) Evolution of the coloration time as a function of cycling for undoped (black trace) and Ti^4+^-doped V_2_O_5_ films (red trace).

**Figure 7 materials-17-04680-f007:**
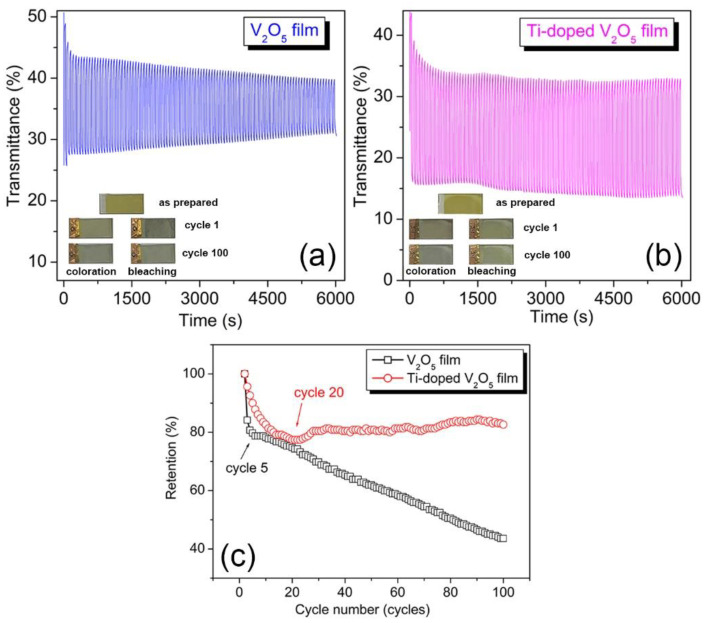
Transmittance measured at 400 nm as a function of time (**a**) for undoped V_2_O_5_; (**b**) for Ti^4+^-doped V_2_O_5_ films. Insets in (**a**,**b**): optical photos of the as-prepared state (top), cycle 1 (middle), and cycle 100 (bottom) for V_2_O_5_ and Ti^4+^-doped V_2_O_5_ films at coloration and bleaching states. (**c**) The evolution of retention as a function of cycling for undoped V_2_O_5_ (black squares) and Ti^4+^-doped V_2_O_5_ films (red circles).

**Table 1 materials-17-04680-t001:** EC optical modulation from the 5th cycle at different voltages of undoped and Ti^4+^-doped V_2_O_5_ films.

	400 nm						
Samples	T_0_ (%) ^¥^	*T_c_* (%) ^¥^	*T_b_* (%) ^¥^	Δ*T* (%) *	$\left|\Delta OD\right|$ ^$^	*Reversibility* ^&^ (%)	CE (cm^2^/C)
Undoped V_2_O_5_ film	28.00	45.50	29.70	−15.80	0.19	93.93	23.57
Ti^4+^-doped V_2_O_5_ film	17.80	35.90	17.80	−18.10	0.30	100.00	34.15

^¥^ T_0_, *T_c_*, *T_b_*: transmittance at 0 V, −3 V, and +3 V, respectively; * Δ*T* (%) = *T_b_ − T_c_*; ^$^ $\left|\Delta OD\right|$: absolute value of ΔOD; ^&^
*reversibility* (%) = 100 × (1−$\left|T_{0}-T_{b}\right|$/$T_{0}$).

## Data Availability

No new data were created or analyzed in this study. Data sharing is not applicable to this article.

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
