# Peer review of "Excellent Electrochromic Properties of Ti4+-Induced Nanowires V2O5 Films"

_materials, 2024, doi:10.3390/ma17194680_

Round 1

Reviewer 1 Report

Comments and Suggestions for Authors

The following issues must be addressed:

1.       Use subscript for all chemical formulas;

2.       The last paragraph from Introduction chapter must outline what is new and innovative in this paper;

3.       Include the solid state reactions corresponding to doping process;

4.       When use the term “Ti ions” include the charge;

5.       What do you mean by “Li deintercalation”?

Author Response

Response to Reviewer 1

Manuscript ID: materials-3150544

Title: Excellent Electrochromic Properties of Ti4+-Induced Nanowires V2O5 Films
Authors: Yufei Deng, Hua Li *, Jian Liang, Jun Liao, Jacques Robichaud, Rui Chen, Yinggui Long, Min Huang, Yahia Djaoued *

Dear Reviewer 1

Thank you for your comments. We responded to them and amended the manuscript according to your suggestions.  

The changes are highlighted in yellow in the annotated manuscript.

Comment 1.  Use subscript for all chemical formulas.

Answer: We have revised all chemical formulas and added the subscripts wherever it was missing in the manuscript.

Comment 2.   The last paragraph from Introduction chapter must outline what is new and innovative in this paper.

Answer: We added two new sentences to explain the purpose of our present work, as seen below:

Lines 63-66

“To our knowledge, this is the first time that the collation between electrochromic properties and impedance of a Ti-induced nanowire film has been dynamically investigated upon cycling. This gives us a better understanding of decaying upon cycling and, therefore, benefits the potential practical application of V2O5-based films.”

Comment 3.  Include the solid state reactions corresponding to doping process.

Answer: We added the solid-state reactions, as seen below:

Lines 166-167

The doping process can be represented by eq. (1).

Comment 4.  When use the term “Ti ions” include the charge.

Answer: We added the charge to read Ti4+-doped V2O5 wherever it occurred in the manuscript.

Comment 5.  What do you mean by “Li deintercalation”?

Answer: “Li deintercalation” refers to Li+ extraction from the lattice.

Reviewer 2 Report

Comments and Suggestions for Authors

please see the attached file. 

Author Response

Response to Reviewer 2

Manuscript ID: materials-3150544

Title: Excellent Electrochromic Properties of Ti4+-Induced Nanowires V2O5 Films
Authors: Yufei Deng, Hua Li *, Jian Liang, Jun Liao, Jacques Robichaud, Rui Chen, Yinggui Long, Min Huang, Yahia Djaoued *

Dear Reviewer 2

Thank you for your comments. We responded to them and amended the manuscript according to your suggestions.  

The changes are highlighted in yellow in the annotated manuscript.

Comment 1.   Lines 25-27 address the issue of poor cycling stability of V2O5, stating, “However, the practical applications of V2O5 in these fields are limited due to its poor cycling stability, which is generally attributed to low electronic conductivity and/or ionic conductivity, slow ion diffusion, and an inert, fragile structure.” Lines 45-46 also discuss this issue, noting that “In the case of V2O5, porous and nanostructured V2O5 still suffer from poor stability due to their intrinsic low conductivity, leading to chemical dissolution and structural fracture.” Both sentences address the problem of poor cycling stability in V2O5. I recommend combining these sentences at the same location for a more cohesive discussion.

Answer: We had difficulty in combining lines 25-27 with lines 45-46 because lines 25-27 address the general limitations of practical applications of V2O5, and lines 45-46 address the specific limitations of porous and nanostructured V2O5; hence, we need to mention again the limitations of porous and nanostructured V2O5 to make a connection between what comes before line 45 with the content of lines 47-50. So, we modified lines 45-46 to read now:

Lines 44-45 of the revised manuscript

“Porous and nanostructured V2O5 suffer from poor stability due to their intrinsic low conductivity.”

Comment 2.   In the Materials and Methods section, some chemicals are listed with their formulas while others are not. Please provide the formulas for all chemicals. Additionally, the grade of ethanol is not specified. Could you include the percentage of ethanol used?

Answer: We added the formulas and grades of the chemicals, as seen below:

Lines 72-77 of the revised manuscript

Vanadium oxytripropoxide (OV(OC3H7)3, VTIP) (98%), Titanium tetraisopropoxide (Ti(OC3H7)4, TTIP) (97%), Lithium perchlorate (LiClO4) (99%) were purchased from Sigma-Aldich LLC. Isopropanol (C3H8O) (99%), acetylacetone (C5H8O2) (99%), acetic acid (C2H4O2) (99%), and ethanol (C2H6O, 99.7%) were purchased from Sinoreagent Co. Ltd. Propylene carbonate (C4H6O3) (99%) was purchased from Ourchem LLC. and Triton X-100 (C34H62O11) was purchased from Macklin LLC.

Comment 3.   Please add subheadings: 2.1 for "Materials" and 2.2 for "Synthesis of V2O5 Precursor Solution."

Answer: We added subheadings 2.1 for "Materials" and 2.2 “for Synthesis of V2O5 Precursor Solution”.                            Comment 4.   In line 98, why is the term "electrochromic layer" used when the abbreviation "EC" for this term has already been introduced?

Answer: We used the abbreviation “EC” wherever “electrochromic” occurred in the manuscript

Comment 5.   In lines 99 and 100, the author has already mentioned "(LiClO4, 99.99%, Sigma-Aldrich)" and "(PC, 99.7%, Sigma-Aldrich)" in the Materials section. Why are these details repeated in this section?

Answer: We removed "(LiClO4, 99.99%, Sigma-Aldrich)" and "(PC, 99.7%, Sigma-Aldrich)" in lines 99 and 100 of the manuscript.

Comment 6.   In lines 101-102, please ensure that all unit formats are consistent (e.g., 2×2 cm²). In line 88, the author used the format “2.5 cm × 2.5 cm.”

Answer: We made the unit format consistent, as seen below:

Line 94 of the revised manuscript

The precursor solutions (36 μl drops) were spread onto 2.5 cm ×2.5 cm ITO substrates.

Line 107 of the revised manuscript

The area of the EC devices was 2.0 cm ×2.0 cm.

Comment 7.   The full names of FE-SEM and TEM are missing. Please provide these.

Answer: The full names of FE-SEM and TEM were added, as seen below:

Line 110, 115

“…using a Field Emission Scanning Electron Microscope (Hitachi S-4800 FE-SEM microscope, Ibaraki-ken, Japan) … Transmission Electron Microscopy (TEM) images were obtained… 

Comment 8.  In lines 114 and 115, the formatting of "V2O5" is incorrect. Please correct it.

Answer: The errors were corrected.

Comment 9.  The details about the platinum grid as a counter electrode and the commercial Ag/AgCl 1 M KCl electrode as a reference, as well as the use of a 0.5 mol/L LiClO4/propylene carbonate solution as an electrolyte, are repeated in lines 116-118 and 123-125. I recommend removing one of these duplicate sections to avoid redundancy.

Answer: The repetition was deleted, and the manuscript was amended as follows:

Lines 128-130:

“Cyclic voltammetry (CV) measurements were performed in the voltage range from −1.5 to 1.5 V at a cycling speed of 100 mV/s, using the same three-electrode cell as in the Electrochemical measurements.”

Comment 10.  What is the thickness of the thin film sample? Have you measured it? Please add the details and the technique used for the measurement.

Answer: We added the thickness of the films and the details and technique used for the measurement as seen below:

Line 111

The thickness of the films was measured from their cross-sectional SEM images using the ImageJ software.

Lines 184-186

The cross-sectional SEM images demonstrate that several nanowires grow from the bottom of the undoped V2O5 film, with the thickness of the bottom part around 110 nm. After doping with Ti4+, the nanowires become dense, with a thickness of around 130 nm.

Comment 11.  Line 127: Please change "electrochromic measurement" to "EC measurement."

Answer: We used the abbreviation “EC” wherever “electrochromic” occurred in the manuscript.

Comment 12.  Line 130: The author mentions the CHI600E electrochemical workstation. Is it the same workstation used for EIS and CV scans in the above section?

Answer: Yes, the CHI600E electrochemical workstation was used for EIS and CV scans. The manuscript was revised, as seen below.

Line 118-127 of the revised manuscript

Electrochemical impedance spectroscopy (EIS) and Cyclic voltammetry (CV) Cyclic voltammetry (CV) were carried out on an electrochemical workstation CHI600E (Chinstruments, Shanghai, China) using a three-electrode cell at open circuit voltage. EIS and CV measurements used the Ti4+-doped V2O5 film (or V2O5 film) deposited on ITO substrates as working electrodes. At the same time, a platinum grid served as a counter electrode, and a commercial Ag/AgCl 1 M KCl electrode served as a reference. A 0.5 mol/L LiClO4 /propylene carbonate solution was used as an electrolyte. The test parameters of EIS were measured in the frequency range from 0.01 Hz to 1000 kHz at the open circuit voltage (OCV) after discharge/charge cycles of 5 mV amplitude. CV measurements were performed in the voltage range from −1.5 to 1.5 V at a cycling speed of 100 mV/s. 

Comment 13.  Why do the authors use +1.5 and -1.5 V for the CV scan, but change to +3 and -3 V for the CA scan? Please explain.

Answer: We used a three-electrode cell with a reference electrode for the CV tests. However, we fabricated an EC device corresponding to a two-electrode cell mode for the EC tests under a CA mode. This difference leads to a higher voltage (at least 3 V), propelling the EC process under a two-electrode cell mode (CA).

Comment 14.  Please add a citation for each equation. I think you did not generate these equations yourself, right?

Answer: Optical density, retention, and coloration efficiency, equations 1, 2, and 3 from lines 133, 141, and 147 of the submitted manuscript, are commonly used quantities or measurements in electrochromic thin film studies, and their origin is seldom, if ever, cited or referenced. However, we added references [28], [29], and [30], but we are not sure they correspond to the original citations.

Comment 15.  Check the format of units throughout the manuscript. For example, lines 131 and 137 use different formats. Choose one and keep it consistent.

Answer: We revised the format of units throughout the manuscript; for instance, line 131 of the submitted manuscript now reads: 

 “… an applied voltage from -3.0 V to +3.0 V, in increasing potential steps of 0.5 V kept for 30 s.”

Comment 16.  Why are the coloration switching behaviour measurements taken at 400 nm? Please explain.

Answer: Switching behavior measurements were taken at 400 nm because the maximum response was observed around this wavelength according to OD results. The corresponding explanation is as seen below:

Lines 263-266

Considering that the largest |OD| occurs at a wavelength of 400 nm for both samples, the following tests for the EC properties, including switching time, coloration efficiency, and cycling stability, were conducted at 400 nm to achieve a strong response.

Comment 17.  Lines 155-156: “The intermediate frequency peaks at ~281, 301, 405, 482, 526, and 700 cm−1 are related to the bending and stretching vibrations (internal modes) of the vanadium-oxide bond in V2_22O5_55.” Can you be more specific about which peaks refer to bending vibrations and which refer to stretching vibrations?

Answer: We modified the text and added a reference to identify the peaks, as seen below:

Line 156-160 of the revised manuscript

The peak at 284 cm−1 is attributed to V=O bending vibration, peaks at 304 and 700 cm−1 correspond to V3-O (triply coordinated oxygen) stretching mode, and peaks at 482 and 526 cm−1 correspond to the bending vibrations of V-O-V (bridging doubly coordinated oxygen) bending vibration. The highest frequency peak at 992 cm-1 corresponds to the stretching mode of the terminal oxygen (vanadyl oxygen, V=On) [32].

Line 477

  1. Su D., Zhao Y., Yan, D., Ding C., Ning M., Zhang J., J. Alloys Compd., 2017, 695, 2974-2980.

Comment 18.  Lines 159-162: The authors claim that TiO2 should appear at 145 and 198 cm−1. However, based on your Raman spectra, it does not show these peaks at all. I only see the peak of V2O5. If the peak does not show, how can the authors make that claim? This means Raman cannot be used to confirm the presence of Ti integrated inside the V2O5.

Answer: In lines 159-162 we wanted to highlight that there is no trace of the TiO2 phase present in the doped film since there are no isolated Raman peaks belonging the TiO2 anatase or rutile phases, demonstrating that Ti4+ exists as a dopant in the V2O5 film. We modified the text, and it now reads:

Line 164 of the revised manuscript

“… demonstrating that no isolated TiO2 phase exists in the Ti4+-doped film and Ti4+ exists as a dopant in the V2O5 lattice. The doping process could be represented by the eq. (4).”

Comment 19.  The quality of the SEM images (a), (c), and (d) needs improvement. They are blurry, and it seems they were taken at low resolution. Why are they still blurry?

Answer: We agree that the SEM images (a), (c), and (d) are blurry, although a professional test center did it. However, we adjusted the contrast and brightness, hoping that the figure is now improved, as seen below:

Figure 2. SEM images of: (a, c, e) V2O5 film; (b, d) Ti4+-doped V2O5 film (nanowires (top) green arrows and nanorod layer (bottom) red arrows). (f, g) Corresponding EDS results for the films in the as-prepared state and after 100 electrochromic cycles (c, d). Insets in (a and b) show the cross-sectional views of the films. Scale bars in (a and b) 1μm, in (c and d) 500 nm, in e 200 nm, inset in (a and b) 500 nm. 

Comment 20.  In the EDS pattern, both samples show similar patterns, and I did not see the peak for Ti as mentioned (I recommend removing). Could you explain why? Normally, EDS has a 5% error margin, and if you claim that the atomic ratio of Ti is 0.03:1, this implies that for every 100 V atoms, there are 3 Ti atoms, which means 3%. This is below the error percentage of EDS. How accurate is this atomic ratio claim in your manuscript? Wavelength-Dispersive X-Ray Spectroscopy (WDS) might be a better solution for detecting low amounts of Ti within the structure. Alternatively, you could use EDS from a TEM to present the atomic ratio. Therefore, consider removing the EDS analysis from the SEM.

Answer: In Fig. 3, we added the EDS spectrum from TEM (see Fig. 3h) to further present the atomic ratio and modified the manuscript accordingly (see Figure 3 h and lines 386-390 below). As for the EDS from SEM, we kept the spectra considering that we compared the pattern before and after cycling for 100 cycles, aiming to understand the changes occurring before and after EC cycling. The manuscript was revised (see lines 385-387 below).

Line 386-390 of the submitted manuscript

The corresponding EDS spectrum (Figure 3h) shows that, after 100 cycles, the atomic ratio Ti:V in the Ti4+-doped V2O5 film is around 0.02:1, close to that of its as-prepared state (0.03:1). In contrast, for the undoped V2O5 film, the content of V ions is too low to be detected in the EDS spectrum, indicating that most of the V2O5 was lost, resulting in correspondingly to high fading upon cycling.

Figure 3. TEM images: (a, b) of a single Ti4+-doped V2O5 nanowire. (c) HRTEM image taken from the square frame in (a) and its diffraction pattern (inset). (d) Retention mapping showing all atoms. Atomic distribution of (e) V, (f) Ti, and (g) O. (h) Corresponding EDS result for the film. 

Comment 21.  Electrochemical impedance spectroscopy (EIS), the authors already mention the full name of this technique in the above section. Please use only EIS here.

Answer: We replaced Electrochemical impedance spectroscopy with its abbreviation “EIS”, as seen below:

Line 208 of the revised manuscript

“EIS was used to understand and analyze the kinetic behavior…”

Comment 22. Suggest adding the HRTEM image of pure V2O5 sample for comparison reason.

Answer: Unfortunately, due to the revision deadline, which required uploading the revised manuscript within 10 days, we don’t have enough time to do HRTEM. 

Comment 23.  What do you mean by "0 cycles"? Do you mean without applying any voltage? Could you also include data for 1 cycle in the Nyquist plot and provide an analysis for it as well?

Answer: Yes, here, 0-cycle refers to the data before cycling. We didn’t test cycle 1. However, we tested cycle 3. The result showed a decline in Rct and KΩ in contrast to cycle 0, consistent with the rule of decay of Rct and KΩ upon cycling. Considering these plots overlap and are hard to see clearly, we didn’t add the plots for cycle 3 in Fig. 4. 

Comment 24.  Improve the discussion on the switching time of both the doped and pure samples. Why does the time increase with the number of cycles in the case of pure sample? Why does the time increase at the 5th cycle and then decrease at the hight number of cycle in the case of doping sample? The current discussion is not clear and convincing enough.

Answer: We added an explanation to the switching time discussion as seen below.

Lines 328-332 of the revised manuscript

“Such an increase in both undoped and Ti4+-doped V2O5 films indicates that at the beginning of the EC process, an activation is needed to adjust the interaction between the electrolyte and electrode. After 5 cycles, the amount of Li+ ions intercalated almost reached saturation, combining a stable structure after the primary activation, and the coloration time for the undoped film got nearly leveled.” 

Comment 25.  Lines 311-312: The Ti-doped V2O5 film has a dense bottom layer, while the undoped V2O5 film features a developed porous structure. Do you have additional evidence to support this claim? The SEM images do not clearly show the porosity of the thin film. Based on your synthesis method, the thin film was calcined at 450 °C. I suppose this temperature may cause agglomeration of the nanowire structure of the thin film, but it does not necessarily mean the film becomes completely dense. The authors recommend providing extra evidence to support this statement. Even films prepared by sputtering techniques can appear smooth in SEM images while having porosity inside the film.

Answer: Presently, we have not found a good alternative to check the porosity, considering the film is too thin to scratch enough V2O5 from the substrate to perform BET or other tests. Yes, the temperature is important. Only a dense film with particles is obtained when we increase the temperature further, such as to 550°C. 

Comment 26.  Lines 334-335: “Such a result is higher than those from the literature for a 2 at.% Mg-doped V2O5 planar film at 50%.” Why compare with Mg-doping? Mg is different from your dopant, which is Ti, and it might behave differently in terms of performance. If the authors wish to compare with the literature, I recommend using Ti-doped V2O5 structures, as I believe some groups have already worked on this before.

Answer: We modified lines 334-335 of the submitted manuscript and added literature to refer only to Ti-doped V2O5 films, as seen below:

Lines 358-362 of the revised manuscript

 “The experimental results we have obtained are significant. Our tested material outperforms those from the literature, which report 65% after 600 cycles for a 4 at.% Ti-doped V2O5 planar film [7] and 50% after 100 cycles for an 8.5 at.% Ti-doped V2O5 film [44], 76% after 100 cycles for a 5% at.% Ti-doped V2O5 film [45], 18.6% for a nanobelt-membrane hybrid-structured vanadium oxide film [17], and 24% for a self-organized multifunctional V2O5 nanofiber-liquid crystal polymer hybrid film [46].”

Line 489-491

  1. Salek G., Bellanger B., Mjejri I., Gaudon M., Rougier A., Inorg. Chem., 2016, 55, 9838–9847.
  2. Wu I-H., Chandrasekar A., Arul K. T., Huang Y.-C., Nga T. T. T., Chen C.-L., Chen J.-L., Wei D.-H., Asokan K., Yeh P.-H., Du C.-H., Chou W.-C., Dong C.-L., Optical Materials:X, 2024, 22, 100301.

Comment 27.  Line 372-373: suggest using atomic ratio from TEM-EDS not from SEM-EDS.

Answer: In Figure 3, we added the EDS spectrum from TEM (Fig. 3(h)) and modified the manuscript accordingly (see lines 388-392) to further present the atomic ratio. Please see our answer to your comment #20.

Comment 28.  Please double check all “electrochromic”word , sometime refer as EC , some time remain as “electrochromic”

Answer: We double-checked the manuscript and changed the word “electrochromic” to “EC” wherever it occurred.

Comment 29.  Please find an alternative reference to replace this 33. Hua S., Jiang N., Chinese Journal of Applied Chemistry, 1992, 9(5), 80–83. (in Chinese). Suggest using English based reference not Chinese reference.

Answer: We replaced this reference with a new one, as seen below:

Line 477

  1. Yu M., Zeng Y., Han Y., Cheng X., Zhao W., Liang C., Tong Y., Tang H., Lu X., Adv. Funct. Mater., 2015, 25, 3534–2540.

Comment 30.  The format of the references is inconsistent. Please double-check them carefully.

Answer: We have revised the reference format to make it consistent.

Reviewer 3 Report

Comments and Suggestions for Authors

The authors studied the electrical and the optical properties of V2O5 films. The authors found that Ti-doping lowered the electrical resistance without deteriorating the optical property. The Ti-doped samples had higher retention than undoped ones. These results highlight that Ti-doping is one of the effective approaches of enhancing the electrical device performances. However, there are some concerns about the introduction and the results. If the authors appropriately revise the manuscript, this study will meet the criteria for the publication in Materials.

Comment list

Comment 1: Why did the authors use vanadium oxytripropoxide as the precursor? There are a lot of precursors of V2O5, such as vanadium acetylacetonate, etc.

Comment 2: Can the authors control the doping amount of Ti? I guess that there is an optimum doping amount.

Comment 3: The distribution of Ti looks non-uniform, as shown in Fig. 3f. I wonder why Ti-doped sample has higher performance than undoped one even if Ti is non-uniformly doped.

Comment 4: In the introduction section, the authors commented on only the application field of V2O5. However, there are a lot of application fields of oxide electronic materials. For example, there are some famous studies about various application fields of oxide materials: ZnO thermoelectric material (ACS Appl. Mater. Interfaces 10, 37709 (2018).), SnO2 chemical sensor (ACS Appl. Mater. Interfaces 6, 357 (2014).), etc. The paper will be improved by commenting on the various application fields and adding the related references including the recommended ones.

Author Response

Response to Reviewer 3

Manuscript ID: materials-3150544

Title: Excellent Electrochromic Properties of Ti4+-Induced Nanowires V2O5 Films
Authors: Yufei Deng, Hua Li *, Jian Liang, Jun Liao, Jacques Robichaud, Rui Chen, Yinggui Long, Min Huang, Yahia Djaoued *

Dear Reviewer 3

Thank you for your comments. We responded to them and amended the manuscript according to your suggestions.  

The changes are highlighted in yellow in the annotated manuscript.

Comment 1. Why did the authors use vanadium oxytripropoxide as the precursor? There are a lot of precursors of V2O5, such as vanadium acetylacetonate, etc.

Answer: To avoid precipitation when mixed with the Ti precursors, we chose vanadium oxytripropoxide as a dissolvable precursor in isopropanol. Vanadium oxytripropoxide has a similar chemical property to titanium tetraisopropoxide. 

Comment 2.   Can the authors control the doping amount of Ti? I guess that there is an optimum doping amount.  

Answer:  We are now trying to control the doping amount of Ti by changing the ratio of precursors, and the corresponding results are being analyzed.

Comment 3.  The distribution of Ti looks non-uniform, as shown in Fig. 3f. I wonder why Ti-doped sample has higher performance than undoped one even if Ti is non-uniformly doped.

Answer: Several reasons contribute to better performance, which we discussed in the article. First, the number of nanowires formed by the Ti4+ dopant minimizes the electric field and contributes to increasing the coloration speed. Second, higher conductivity after doping leads to minimized polarization and consequently to improved cycling stability

Comment 4.  In the introduction section, the authors commented on only the application field of V2O5. However, there are a lot of application fields of oxide electronic materials. For example, there are some famous studies about various application fields of oxide materials: ZnO thermoelectric material (ACS Appl. Mater. Interfaces 10, 37709 (2018).), SnO2 chemical sensor (ACS Appl. Mater. Interfaces 6, 357 (2014).), etc. The paper will be improved by commenting on the various application fields and adding the related references including the recommended ones.

Answer:   We thank the reviewer for suggesting we include references about other oxide electronic materials like ZnO and SnO2. However, we decided not to include them since the present study focuses on the EC properties of Ti-dopped V2O5, and we think these references would not significantly enhance our manuscript. Furthermore, adding the references would mean enlarging the discussion to other oxide materials.

Round 2

Reviewer 1 Report

Comments and Suggestions for Authors

The manuscript can be published in present form.

Author Response

Response to Reviewer 1 

Manuscript ID: materials-3150544R1

Title: Excellent Electrochromic Properties of Ti4+-Induced Nanowires V2O5 Films
Authors: Yufei Deng, Hua Li *, Jian Liang, Jun Liao, Jacques Robichaud, Rui Chen, Yinggui Long, Min Huang, Yahia Djaoued *

Comments and Suggestions for Authors

The manuscript can be published in present form.

Answer:

We sincerely thank you for your careful review and valuable suggestions on our manuscript.

Reviewer 3 Report

Comments and Suggestions for Authors

The authors addressed my concern (comment 1) according to my comments. However, the authors’ answers to comment 2-4 are unclear. I ask the authors 3 questions including my previous comments.

Comment 2: Can the authors control the doping amount of Ti? I guess that there is an optimum doping amount. Please show the optimum amount. If the authors are now trying to control, please submit the paper after the experiment of Ti amount optimization. This is important because Ti doing is originality.

Comment 3: The distribution of Ti looks non-uniform, as shown in Fig. 3f. I wonder why Ti doped sample has higher performance than undoped one even if Ti is non-uniformly doped. Although the authors replied to my comment, the reply did not clarify the influence of non-uniformity on the performance.

Comment 4: In the previous review, we made a comment as follows. In the introduction section, the authors commented on only the application field of V2O5. However, there are a lot of application fields of oxide electronic materials. For example, there are some famous studies about various application fields of oxide materials: ZnO thermoelectric material (ACS Appl. Mater. Interfaces 10, 37709 (2018).), SnO2 chemical sensor (ACS Appl. Mater. Interfaces 6, 357 (2014).), etc. The paper will be improved by commenting on the various application fields and adding the related references including the recommended ones.

The authors replied to my comment as follows: “…However, we decided not to include them since the present study focuses on the EC properties of Ti-dopped V2O5, and we think these references would not significantly enhance our manuscript….” Because there are a lot of electronic oxide materials except for V2O5, my comment is important. In general, if the authors do not revise the manuscript only based on the authors’ own idea, there is no meaning of review process. Specifically, ZnO is historically more popular oxide material than V2O5. Please reply to my comments faithfully. The reviewers are spending a lot of time on the review process.

Author Response

Response to Reviewer 3 

Manuscript ID: materials-3150544R1

Title: Excellent Electrochromic Properties of Ti4+-Induced Nanowires V2O5 Films
Authors: Yufei Deng, Hua Li *, Jian Liang, Jun Liao, Jacques Robichaud, Rui Chen, Yinggui Long, Min Huang, Yahia Djaoued *

Dear Reviewer 3

Thank you for your comments. We responded to them and amended the manuscript according to your suggestions.  

The changes are highlighted in yellow in the annotated manuscript.

Comments and Suggestions for Authors

The authors addressed my concern (comment 1) according to my comments. However, the authors’ answers to comment 2-4 are unclear. I ask the authors 3 questions including my previous comments.

Comment 2: Can the authors control the doping amount of Ti? I guess that there is an optimum doping amount. Please show the optimum amount. If the authors are now trying to control, please submit the paper after the experiment of Ti amount optimization. This is important because Ti doing is originality.

Answer:

Indeed, it is possible to control doping, as evidenced by the results obtained from preliminary experiments, which have identified the optimal doping level. In that work, we kept the quantity of vanadium precursor constant while varying the amount of titanium precursor. We thus prepared five samples, which resulted in different atomic ratios of titanium to vanadium (0:1, 0.03:1, 0.06:1, 0,11:1, and 0.30:1). The comparison of the optical contrast at 400 nm for those different samples demonstrated the advantage of the sample with a 0.03:1 ratio showing an optical contrast of 34.7%, which is higher than for the other samples (30.7% for 0:1; 26.5% for 0.06:1, 18% for 0.11:1 and 24.4 for 0.30:1). Therefore, in the present work, we further used samples with the optimal ratio (0.03:1) to compare their electrochromic properties with the undoped sample with an atomic ratio of titanium to vanadium of 0:1. 

Comment 3: The distribution of Ti looks non-uniform, as shown in Fig. 3f. I wonder why Ti doped sample has higher performance than undoped one even if Ti is non-uniformly doped. Although the authors replied to my comment, the reply did not clarify the influence of non-uniformity on the performance.

Answer:

As for the dopant non-uniformity on the performance, Figure 3 (d to g) shows the atomic distribution of the elements on a single nanorod. Figure 3(f) shows a uniform distribution of Ti except for a tiny part in the middle of the nanorod (see figure below), where Ti is more densely packed. The EC device contains a multitude of nanorods that, unfortunately, cannot be probed individually. Considering the whole EC device, we think the Ti distribution can be regarded as relatively uniform. As we mentioned in the article, several factors contribute to better performance. First, the specific morphology, i.e., nanowire induced by Ti4+ dopant, minimizes the electric field and contributes to increased coloration speed; second, higher conductivity after doping leads to minimized polarization and improved cycling stability. Consequently, the results prove that the increasing coloration speed is achieved due to doping with Ti4+.

Figure 3f: Atomic distribution of Ti.

Comment 4: In the previous review, we made a comment as follows. In the introduction section, the authors commented on only the application field of V2O5. However, there are a lot of application fields of oxide electronic materials. For example, there are some famous studies about various application fields of oxide materials: ZnO thermoelectric material (ACS Appl. Mater. Interfaces 10, 37709 (2018).), SnO2 chemical sensor (ACS Appl. Mater. Interfaces 6, 357 (2014).), etc. The paper will be improved by commenting on the various application fields and adding the related references including the recommended ones.

The authors replied to my comment as follows: “…However, we decided not to include them since the present study focuses on the EC properties of Ti-dopped V2O5, and we think these references would not significantly enhance our manuscript….” Because there are a lot of electronic oxide materials except for V2O5, my comment is important. In general, if the authors do not revise the manuscript only based on the authors’ own idea, there is no meaning of review process. Specifically, ZnO is historically more popular oxide material than V2O5. Please reply to my comments faithfully. The reviewers are spending a lot of time on the review process.

Answer:

We added a new paragraph at the beginning of the introduction to better consider the various application fields of oxide materials. We added two more references as well as the references suggested by the reviewer, as seen below:

Lines 24-28 (revised manuscript)

“Transition metal oxides (TMOs) offer a wide range of applications for energy storage and energy conversion [1], waste treatment [2], and gas sensing, to name a few. For instance, Ishibe et al. showed the power enhancement of embedded-ZnO nanowire structure for energy conversion [3], while Maeng et al. presented a highly sensitive SnO2 nano slab for NO2 gas sensing [4].”

We also amended the second paragraph to now read:

Lines 29-32 (revised manuscript)

“Among TMOs, V2O5, owing to its layered structure and high Lithium-ion intercalation capacity, has attracted considerable attention for promising applications in devices such as batteries [5,6], supercapacitors [7], and electrochromic (EC) smart windows [8]. However…”

The reference numbers in the manuscript and the References section were modified accordingly.

Lines 447-452 (revised manuscript)

  1. Yuana , Duana X., Liua J., Yun Ye Y., Lva F, Liu T., Wang Q., Zhang X., Energy Storage Materials, 42, 2021, 42, 317–369.
  2. Okpara C., Olatunde O. C., Wojuola O. , Onwudiwe D. C., Environmental Advances, 2023, 11, 100341.
  3. Ishibe T., Tomeda A., Watanabe K., Kamakura Y., Mori N., Naruse N., Mera Y., Yamashita Y., Nakamura Y., ACS Appl. Interfaces,2018, 10, 43, 37709–37716.
  4. Maeng S., Kim S.-W., Lee D.-H., Moon S.-E., Kim K.-C., Maiti A., ACS Appl. Interfaces,2014, 6, 1, 357–363.
